# Evaluation of the Consistency of Three GRACE Gap-Filling Data

An Qian [1,2,3], Shuang Yi [4,*], Feng Li [1,3], Boli Su [5], Guangtong Sun [1,6] and Xiaoyang Liu [1]

1   School of Ecology and Environment, Institute of Disaster Prevention, Sanhe 065201, China
2   Hebei Key Laboratory of Earthquake Dynamics, Institute of Disaster Prevention, Sanhe 065201, China
3   Hebei Key Laboratory of Earthquake Disaster Prevention and Risk Assessment, Institute of Disaster Prevention, Sanhe 065201, China
4   Key Laboratory of Computational Geodynamics, University of Chinese Academy of Sciences, Beijing 100049, China
5   School of Electronics and Control Engineering, Institute of Disaster Prevention, Sanhe 065201, China
6   Beijing Key Laboratory of Urban Spatial Information Engineering, Beijing 100038, China
*   Correspondence: s.yi@ucas.ac.cn; Tel.: +86-010-8825-6237

**Abstract:** The Gravity Recovery and Climate Experiment (GRACE) gravity mission has become a leading platform for monitoring temporal changes in the Earth's global gravity field. However, the usability of GRACE data is severely limited by 11 months of missing data between the GRACE and GRACE Follow-on (GRACE-FO) missions. To date, several approaches have been proposed to fill this data gap in the form of spherical harmonic coefficients (an expression of the Earth's gravity field, SHCs). However, systematic analysis to reveal the characteristics and consistency of the datasets produced by these latest gap-filling techniques is yet to be carried out. Here, three SHC gap-filling products are systematically analyzed and compared: (1) Combining high–low satellite-to-satellite tracking with satellite laser ranging (SLR) observations (QuantumFrontiers, QF), (2) SLR-based recovery incorporating the GRACE empirical orthogonal function decomposition model proposed by the Institute of Geodesy and Geoinformation at the University of Bonn (hereafter, denoted as IGG), and (3) applying the singular spectrum analysis approach (SSA). The results show that (1) the SHCs of the QF, IGG, and SSA data are consistent up to degree 12; (2) the IGG and SSA data give similar results over the 11 gap months, but the IGG shows a faster increase in the mean ocean water mass and the SSA appears to better capture the interannual variation in the terrestrial water storage; and (3) the noise level increases significantly in the high-degree terms ($l > 16$) of the QF data, so these data are only applicable for large-scale mass migration research. These results provide a reference for users to select a gap-filling product. Finally, we propose a new scheme based on the triple collocation method to derive a weight matrix to fuse these three datasets into a more robust solution.

**Keywords:** satellite gravity; time variable gravity; time-series analysis; spatial analysis; sea level change; gap-filling; triple collocation

## 1. Introduction

The earth's time-varying gravity field mainly reflects the redistribution of mass in the global water cycle system and within the earth. Studying the changes in the gravity field helps us to understand the earth's internal structure and monitor the global environment and climate changes. Traditional gravity data are derived from ground-based gravimetry, satellite altimetry data in ocean areas [1], and satellite orbit data in terms of the expansion coefficients of the spherical function [2]. However, the limited density, temporal resolution, and spatial coverage of these data prevent an in-depth exploration of the physics of the Earth's interior and the interactions among land water storage, glaciers, and oceans. The successful launch of the GRACE gravity satellites in 2002, jointly developed by the US and Germany, made it possible to monitor time-varying changes in the global gravity

field with unprecedented spatial and temporal resolution [3]. The gravity field products provided by GRACE gravity satellites have been widely used and greatly contributed to our understanding of seismic activities, terrestrial water storage, sea level changes, and polar and mountain ice melt [4–6].

The GRACE gravity satellites provide standard scientific products of Earth's monthly mean gravity field in the form of spherical harmonic coefficients (SHCs) resolved up to degrees 60 or 96. The main GRACE data processing agencies consist of the Center for Space Research (CSR) at the University of Texas at Austin, the German Research Centre for Geosciences, and the NASA Jet Propulsion Laboratory. Although the designed lifetime of the GRACE gravity satellite was only five years, the satellite has fulfilled its scientific mission, and its service was extended to June 2017. GRACE's successor satellite, GRACE-FO (GFO), was launched in May 2018 and began to deliver time-varying gravity products one month later. GFO continues to carry out the GRACE science mission, but the 11-month gap between the two missions inevitably limits our ability to systematically analyze and fully utilize the satellite observations of GRACE and GFO over the past two decades.

The missing one or two months of data during the service life of GRACE are usually filled by interpolation processes, such as cubic spline interpolation [7,8]. However, filling the 11-month gap between GRACE and GFO presents a major challenge. Gravity observations from other satellites provide promising solutions. For example, laser ranging technology (SLR) has been used to track long-term changes in the earth's gravitational field since the 1970s, when the Starlette satellite of the French National Center for Space Research and the LAGEOS satellite of NASA were deployed [9]. Although SLR yields lower-resolution gravity fields than GRACE, it has a longer operational lifetime and continues even when GRACE missions are disrupted, opening up the possibility of using SLR data to fill in the gaps between GRACE and GFO [10–12]. Unfortunately, this approach yields only 10-degree gravity field models, which capture only the long-term variability of the large-scale gravity fields [12]. Furthermore, the long-wave variations in the time-varying gravity field can also be monitored by high–low satellite-to-satellite tracking (HLSST) measurements, but this technique provides only the long-wave variations in the gravity field with an approximate spatial resolution of 2000 km [13]. Similarly, orbit data from the geomagnetic satellite constellation Swarm is also able to solve the Earth's large-scale time-varying gravity field [14], but its data quality is only good at low degrees (i.e., below degree 12 [15]).

To overcome the low-resolution problems of SLR and HLSST, researchers have proposed various techniques for recovering the time-varying gravity field and filling the 11-month gap between the GRACE and GFO data. Empirical orthogonal decomposition methods [16], machine learning techniques [17–20], and singular spectrum analysis [21] essentially maintain the spatial resolution of GRACE. The empirical orthogonal function (EOF) analysis method is a way of dimensionality reduction of data to obtain the dominant signal [22]. Except for SLR alone, a time-varying gravity field can be recovered from SLR data using the GRACE EOF decomposition model [16]. Machine learning is a popular branch of artificial intelligence, which is now widely used for solving engineering and science problems. However, machine learning is usually designed for regional gridded observations of hydrological signals, rather than for global and generic SHCs. Singular spectrum analysis methods, which have recently emerged for studying non-linear time series data, promise the availability of simple and generous models with less-computationally intensive solutions and high operability. This approach decomposes a time series into its trend, periodic components, and noise [23,24], and it opens up an effective way to fill the gaps between GRACE and GFO. The first attempt to fill data gaps with SSA data [25] was targeted at regional terrestrial water storage anomalies in China and was not generalizable to other research fields (i.e., oceanography and solid Earth physics). Yi and Sneeuw (2021) [21] and Wang et al. (2021) [26] improved the existing work by devising new schemes to fill the data gaps with the highly applicable form of SHCs.

In general, there are three types of SHC gap-filling datasets: Purely derived from other observations but with limited resolution, fully data-driven approaches that inherit information from GRACE/GFO observations, and hybrid approaches integrating SLR or other data with GRACE. Here, we choose each instance of these three types for comparison: (1) Time-varying gravity field models recovered from a combination of HLSST and SLR (QF, Weigelt 2019 [27]), (2) interpolations of GRACE observations using the SSA approach (Yi and Sneeuw 2021 [21]), and (3) a hybrid modelling that using EOFs from the GRACE solutions as the base functions to recover time-varying gravity field models from SLR observations (IGG, Löcher & Kusche 2021 [16]). In this paper, we will scrutinize these three datasets in the spectral and spatial domains, evaluate their consistency, and explore their applicable scenarios.

## 2. Materials and Methods

### 2.1. GRACE Products and Water Storage Estimates

In this study, we investigated three GRACE gap-filling products (SSA, IGG, and QF) in the form of SHCs. Table 1 lists the satellites from which these data were derived. The SSA data were based on the GSM RL06 level-2 product from CSR truncated to degree 60, and the time span is from January 2003 to December 2020 [21]. The degree-1 terms were supplemented using the method of Sun et al. (2016) [28], and the C20 term was replaced by the result of SLR [29].

**Table 1.** The HLSST LEO satellites, SLR, and low-low satellite-to-satellite tracking (LLSST) satellites used for obtaining the SSA, IGG, and QF data.

| Data Style | Satellite | Satellite Type |
|---|---|---|
| SSA | GRACE, GRACE-FO | Gravity |
| IGG | Lageos ½, AJISAI, Starlette, Stella | Geodetic SLR |
| | GRACE, GRACE-FO | Gravity |
| QF | Champ, GRACE A/B, GOCE | Gravity |
| | Swarm A, B, C | Geomagnetic |
| | TanDEM-X, TerraSAR-X, Kompsat5, Sentinel 1A, 1B, 2A, 3A | SAR |
| | SAC-C, CNOFS | Environmental monitoring |
| | Cosmic 1-6, MetOpA, MetOpB | Weather |
| | Jason 1-3 | Altimetry |
| | Lageos ½, LARES, Starlette, Stella, Larets, AJISAI, Beacon-C, Blits | Geodetic SLR |

The IGG data are truncated to degree 60 and the data time span is from January 2003 to June 2020 [16]. It was solved from SLR observations using empirical orthogonal functions (EOFs) from the GRACE solutions as the base functions. The time-varying gravity field of ITSG-Grace2018 (155 months, April 2002–August 2016 [30]) was used as a source of EOFs after replacing C20 by the result of SLR [29]. To mitigate the truncation effect of EOFs and ensure the stationarity of the signals outside the GRACE time frame, the high-degree and low-degree terms of the SHCs are estimated separately: The high-degree terms are estimated based on the leading modes in the EOF decomposition of the GRACE solutions, while the low-degree terms are inverted from the SLR observations [16]. Considering the different maximum degrees of SLR solutions estimated in IGG, Löcher and Kusche (2021) provided six combinations of time-varying gravity field models [16]. The first five of these models are listed in Table 2, and the sixth model is their weighted average (EnsMean). As mentioned in Löcher and Kusche (2021), whether focusing on the global or on regions with spatially homogeneous signals, EnsMean is a good compromise of the first five models, except for some limitations in Greenland [16]. Therefore, we adopt EnsMean hereafter.

**Table 2.** Solution types of the IGG data [16].

| Types | SLR Degrees Estimated | GRACE EOFs Applied | GRACE Degrees Used in EOFs |
|-------|----------------------|--------------------|----------------------------|
| S0+6E | None | 6 | 2–60 |
| S2+6E | 2 | 6 | 3–60 |
| S3+6E | 2–3 | 6 | 4–60 |
| S4+6E | 2–4 | 6 | 5–60 |
| S5+4E | 2–5 | 4 | 6–60 |

The QF data constitute a series of monthly gravity field models with SHCs up to degree-and-order (d/o) 60. The data are based on HLSST tracking and SLR data provided by the International Centre for Global Earth Models. Nine SLR geodetic satellites and 27 low-earth orbiting (LEO) satellites are utilized in the computation, including four dedicated gravity satellites: CHAMP, GOCE, and GRACE A/B [27]. Details of these satellites are given in Table 1. The SLR and HLSST data are combined at the normal equation level using Variance Component Estimation, and their spatial resolution is approximately 1000–2000 km [13,31]. The v2-Kalman filtered QF data, provided by Weigelt (2019) [27], was used in this study, and the data time span is from January 2003 to December 2018.

As the SHCs of the IGG and QF data do not contain degree-1 terms, we supplement the degree-1 terms of IGG and QF with these from Sun et al. (2016) (same as SSA data) [28]. In addition, we first filtered these SHCs using the DDK5 method [32], and then expressed them in the form of equivalent water height (EWH) [33]:

$$\Delta h = \frac{a\rho_e}{3\rho_w} \sum_{l=0}^{60} \sum_{m=0}^{l} \frac{2l+1}{1+k_l} \overline{P}_{lm}(\cos\theta)(\Delta\overline{C}_{lm}\cos m\lambda + \Delta\overline{S}_{lm}\sin m\lambda) \tag{1}$$

where $\Delta h$ is the variation of EWH, $\rho_e$ and $a$ are the mean density and radius of the Earth, respectively, $\rho_w$ is the water density, $\theta$ and $\lambda$ are the geocentric colatitude and geocentric longitude, respectively, $l$ and $m$ represent the degree and order of the SHCs, respectively, $k_l$ is the load Love number at degree $l$ [34], $\overline{P}_{lm}$ is the fully normalized Legendre function of degree l and order m, and $\Delta\overline{C}_{lm}$ and $\Delta\overline{S}_{lm}$ are the anomalies of the fully normalized SHCs.

*2.2. Assessment Indicators*

The consistency of the time series was evaluated using the root mean square (RMS) and relative RMS (rRMS) values. The RMS evaluates the variability of a time series of N samples, which represents signal strength, whereas the rRMS assesses the agreement between two series. The smaller the rRMS value between the two series, the higher the consistency. When rRMS = 0, the data are identical; when rRMS = 1, it means that the residual variance has not changed, that is, the complexity of the data has not been reduced, indicating the data are uncorrelated. We also calculated the correlation between the two series. In general, when two independent time series are affected by complex and variable factors, they can be considered strongly correlated if their correlation coefficient exceeds 0.5 subject to the premised significance [35]. Therefore, in the comparative analysis of the SSA, IGG, and QF data, a good correlation between two sets of time series was assumed if the correlation coefficient was over 0.5. The RMS, rRMS, and correlation coefficient are, respectively, calculated as

$$\mathrm{RMS}(x) = \sqrt{\frac{1}{N}\sum_{i=1}^{N} x_i^2} \tag{2}$$

$$\mathrm{rRMS}(x,y) = \mathrm{RMS}(x-y)/\mathrm{RMS}(y) \tag{3}$$

$$\rho_{xy} = \frac{\sum\limits_{i=1}^{N}(x_i - \overline{x})(y_i - \overline{y})}{\sqrt{\sum\limits_{i=1}^{N}(x_i - \overline{x})^2 \cdot \sum\limits_{i=1}^{N}(y_i - \overline{y})^2}} \tag{4}$$

where $x$ and $y$ represent two series, $N$ represents the number of samples, and $\bar{x}$ and $\bar{y}$ are the sample averages of $x$ and $y$, respectively.

### 2.3. Fusion Method Based on Triple-Collocation

SSA, IGG, and QF all have the ability to fill in the missing data of GRACE, but we cannot treat any of them as a genuine observation of the "truth" due to the uncertainties in their data processing methods. In the absence of significant coarse errors in observations, the weighted average result can be considered the optimal estimate of the missing GRACE data. Here we introduced a triple configuration (TC) approach to fuse these three gap-filling data. The TC method was firstly proposed by Stoffelen to correct for scattermeter-driven ocean wind speed and estimate its error [36]. It has been widely used for estimating random error variances and uncertainties in remote sensing data, such as soil moisture [37,38], leaf area index [39], sea surface salinity [40], and ocean winds [36,41].

The key to the fusion of SSA, IGG, and QF data is the determination of the weights, which are closely related to the variance of the observations. The TC method can be a promising candidate for estimating the error variance of SHCs of SSA, IGG, and QF. Assuming that the errors of these data are independent of each other, a linear regression model is constructed:

$$X_i = \alpha_i + \beta_i t + \varepsilon_i \tag{5}$$

where $X_i$ ($i$ = 1, 2, 3) represents the SHCs series of the SSA, QF, and IGG, respectively. $t$ represents the true underlying value. $\varepsilon_i$ represents the error of SHCs of SSA, QF, and IGG, respectively. $\alpha_i$ and $\beta_i$ are the intercept and slope, respectively.

Assuming that the error expectations for these three data sources are zero and uncorrelated with each other and with $t$, we can obtain the TC estimation equation for error variances:

$$\begin{bmatrix} \sigma_{\varepsilon_1}^2 \\ \sigma_{\varepsilon_2}^2 \\ \sigma_{\varepsilon_3}^2 \end{bmatrix} = \begin{bmatrix} Q_{11} - \dfrac{Q_{12}Q_{13}}{Q_{23}} \\ Q_{22} - \dfrac{Q_{12}Q_{23}}{Q_{13}} \\ Q_{33} - \dfrac{Q_{13}Q_{23}}{Q_{12}} \end{bmatrix} \tag{6}$$

where $Q_{11}$, $Q_{12}$, $Q_{13}$, $Q_{22}$, $Q_{23}$, and $Q_{33}$ are the covariances. $\sigma_{\varepsilon_1}^2$, $\sigma_{\varepsilon_2}^2$, and $\sigma_{\varepsilon_3}^2$ are the error variances of SSA, QF, and IGG, respectively. A detailed derivation process could refer to McColl et al. (2014) [42].

Then we can use Equation (7) to incorporate these three data.

$$Y = \varpi_1 X_1 + \varpi_2 X_2 + \varpi_3 X_3 \tag{7}$$

where $Y$ is the result after fusion, and $\omega_i$ represents weight calculated from

$$\begin{bmatrix} \varpi_1 \\ \varpi_2 \\ \varpi_3 \end{bmatrix} = \begin{bmatrix} \dfrac{\delta_{\varepsilon_2}^2 \delta_{\varepsilon_3}^2}{\delta_{\varepsilon_1}^2 \delta_{\varepsilon_2}^2 + \delta_{\varepsilon_2}^2 \delta_{\varepsilon_3}^2 + \delta_{\varepsilon_3}^2 \delta_{\varepsilon_1}^2} \\ \dfrac{\delta_{\varepsilon_3}^2 \delta_{\varepsilon_1}^2}{\delta_{\varepsilon_1}^2 \delta_{\varepsilon_2}^2 + \delta_{\varepsilon_2}^2 \delta_{\varepsilon_3}^2 + \delta_{\varepsilon_3}^2 \delta_{\varepsilon_1}^2} \\ \dfrac{\delta_{\varepsilon_1}^2 \delta_{\varepsilon_2}^2}{\delta_{\varepsilon_1}^2 \delta_{\varepsilon_2}^2 + \delta_{\varepsilon_2}^2 \delta_{\varepsilon_3}^2 + \delta_{\varepsilon_3}^2 \delta_{\varepsilon_1}^2} \end{bmatrix} \tag{8}$$

## 3. Results and Analysis

### 3.1. Comparative Analysis in the Spectral Domain

First, we scrutinize these three SHC datasets in the spectral domains over three periods: The end of GRACE (January 2016–June 2017), the gap months between the GRACE and GFO missions (July 2017–May 2018), and the beginning of GFO (June 2018–December 2018).

The Swarm solutions (http://icgem.gfz-potsdam.de/series/02_COST-G/Swarm, Potsdam, German, April 2022) for the same period are used as a reference for the calculation of rRMS values Equation (3). The average gravity field measured from GRACE for the period January 2003 to December 2010 is subtracted from the Swarm result. Considering the limited spatial resolution of the QF, only the low-degree terms (degree ≤11,140 coefficients in total) were compared.

Figure 1 plots the rRMS values between the three datasets and Swarm solutions on the degree-order plane. The colored squares represent rRMS values and the white ones indicate the rRMS values are higher than 1.0. The numbers of coefficients with rRMS values between 0.5 and 1.0 and within the range <0.5 are enumerated separately and listed in the upper left corner of each subplot. According to Figure 1, the statistics of rRMS values for QF and SSA are largely comparable across the three periods. There are approximately 82% of rRMS values smaller than 1 and approximately 25% smaller than 0.5. In contrast, at the end of GRACE, the statistics of rRMS values for IGG are comparable to the other two datasets. However, in the early GFO and the gap periods, only approximately 75% of IGG's rRMS values are less than 1, a bit lower than the other two datasets. The main reason for this discrepancy may be that the IGG recovers the time-varying gravity fields from the SLR using only GRACE data from February 2002 to August 2016 [16]. Note that both the QF time-varying gravity field solutions also incorporate the Swarm solutions (Table 1). SSA is a purely data-driven approach that fully inherits from GRACE/GFO observations, so its result generally reflects the consistency between GRACE and Swarm datasets.

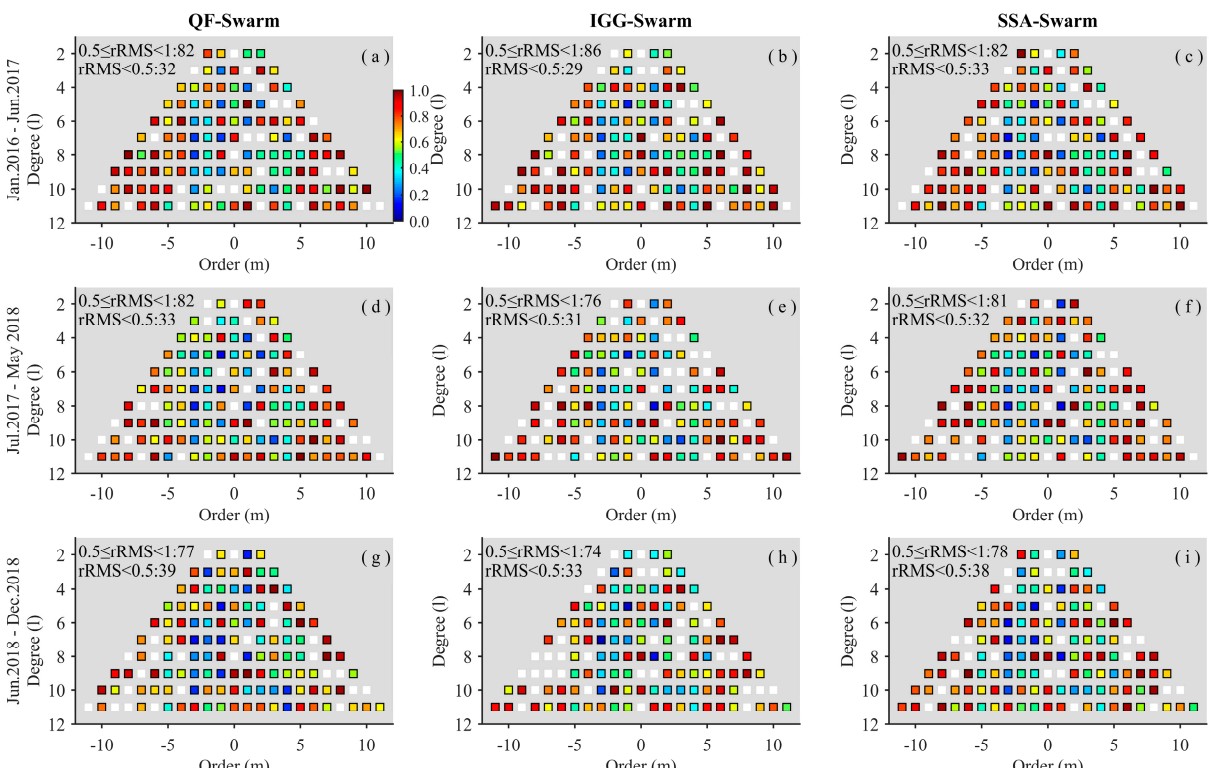

**Figure 1.** rRMS values of the SHCs on the d/o plane (2 ≤ l ≤ 11) between the three datasets and Swarm datasets over three time periods: (**a**–**c**) End of GRACE (January 2016–June 2017), (**d**–**f**) the 11-month gap (July 2017–May 2018), and (**g**–**i**) beginning of GFO (June 2018–December 2018). The number of rRMS values in different intervals is annotated. The color scale is shown in subplot a.

It is worth noting that there may be certain differences among SSA, IGG, QF, and Swarm coefficients with rRMS values less than 1. Figure 2 shows some examples in detail, including consistent or inconsistent series. The values on the top left are the RMS values for IGG, SSA, QF, and Swarm, respectively. As shown in Figure 2, the RMS values for

the coefficients of $C_{3,0}$, $S_{3,1}$, $C_{5,0}$, and $C_{6,0}$ are roughly comparable. Although the Swarm SHCs time series exhibits higher volatility than the others, especially before mid-2015, the SHCs of IGG, SSA, QF, and Swarm are essentially consistent on average. The consistency deteriorates when the seasonality is weak, e.g., $C_{2,1}$ and $C_{4,0}$. Another possible reason for the discrepancies is, as discussed in Teixeira da Encarnação et al. (2020) [15], the different mean pole models [43] used between GRACE and Swarm solutions. Furthermore, compared to QF, it is noticeable that there is better consistency between IGG and SSA (Figure 2a,d). The main reason for this may be that both SSA and IGG data are highly inherited from GRACE/GFO.

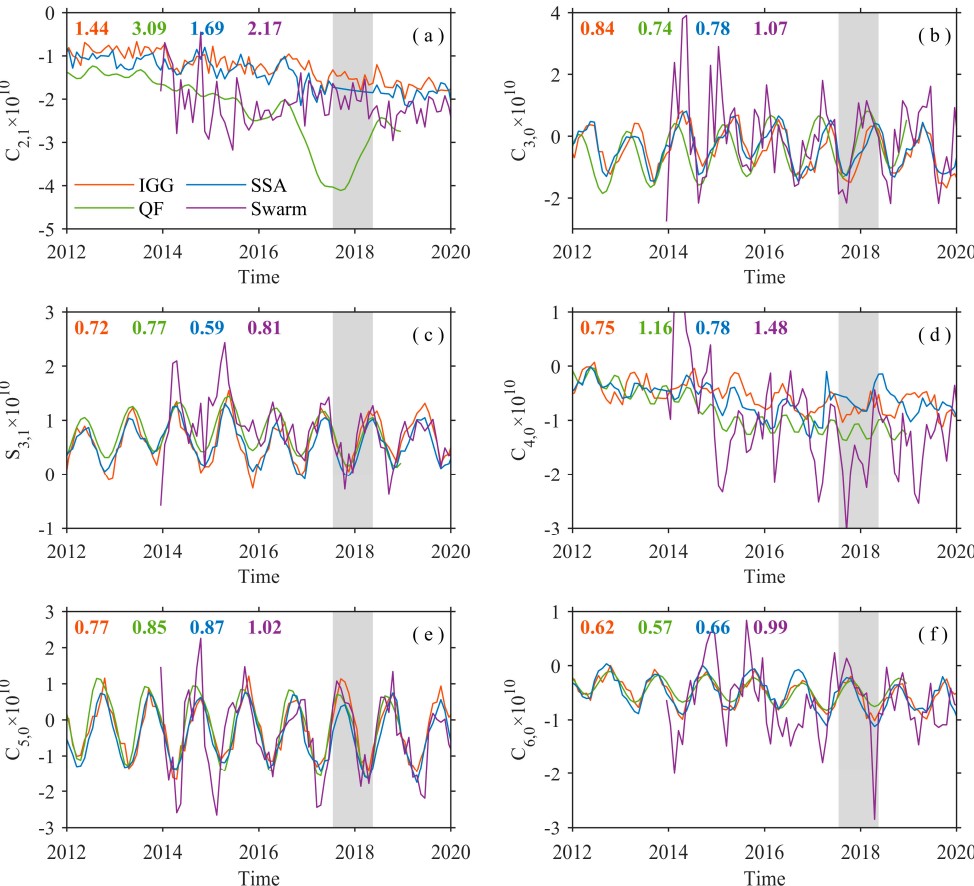

**Figure 2.** Time series of selected coefficients of the IGG, SSA, QF, and Swarm data: (**a**) $C_{2,1}$, (**b**) $C_{3,0}$, (**c**) $S_{3,1}$, (**d**) $C_{4,0}$, (**e**) $C_{5,0}$, (**f**) $C_{6,0}$. The upper left values are the RMS values for IGG, SSA, QF, and Swarm, respectively.

Next, we calculated the degree variance of the SHCs (degree 5–60) of the IGG, QF, and SSA data during the 11-month gap (July 2017 to May 2018) between GRACE and GFO (Figure 3), which illustrates the signal intensity of each degree term of the SHCs. The variances of the SHCs of the IGG and SSA data gradually decrease with an increasing degree and show a fairly consistent trend. The variances of the lower-degree terms of the QF coefficients ($l \leq 12$–16) are essentially consistent with those of SSA and IGG, but the variances of the higher-degree terms of QF progressively increase with an increasing degree, indicating a rapid increase in the noise level. This phenomenon is mainly attributable to the limited observational techniques and spatial resolution of QF [13,31]. Therefore, with spatial resolution accuracy much lower than GRACE, QF is only suitable for monitoring large-scale mass transport (1000–2000 km). The degree variance in IGG is slightly larger than in SSA, especially for coefficients with degrees > 40 and for the September 2017 result (Figure 3c). In other words, the signal intensity in SSA is slightly smaller than in IGG.

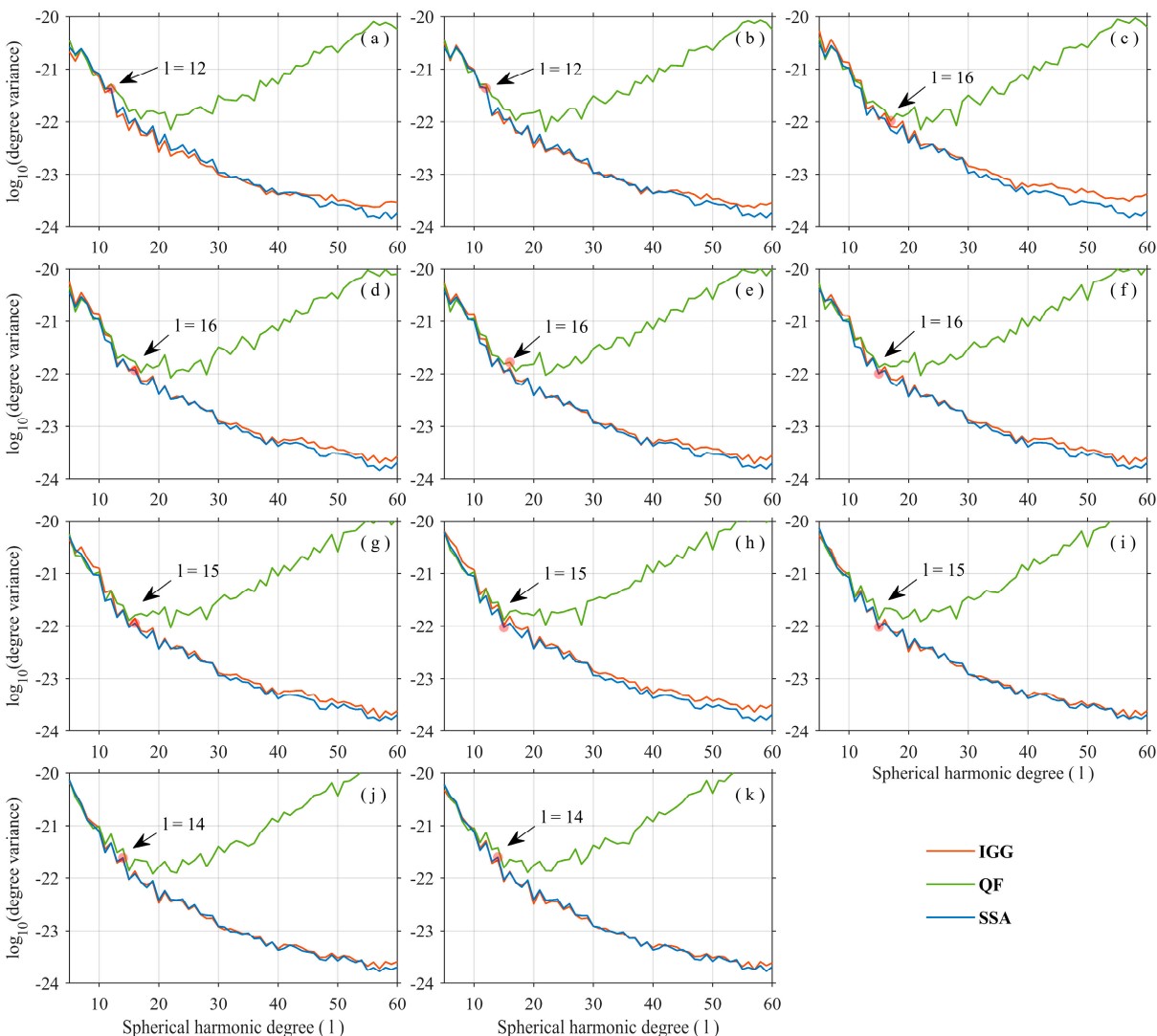

**Figure 3.** Logarithmic (base 10) variances versus degree of SHCs of the IGG, QF, and SSA data during the 11-month gap (July 2017–May 2018). (**a**–**k**) denote the time series from July 2017 to May 2018, respectively.

### 3.2. View of EWH Results of SHCs in the Space Domain

In the section, we give a perspective on the characteristics of different datasets when applied to study miscellaneous signals. The five sites given in Figure 4f cover three categories: Polar ice melting in Greenland (Figure 4a) and Antarctica (Figure 4b), terrestrial water storage changes in north India (Figure 4c) and the Amazon (Figure 4d), and seismic activity in eastern Japan (Figure 4e). Two types of EWH results were calculated from SHCs: EWH with d/o up to 60 and EWH with d/o up to 12 (to account for the limited spatial resolution of QF). As the effect of glacial isostatic adjustment (GIA) affects only linear trends, it was not considered here. It should be noted that the SSA data used the original GRACE observations for the period outside the gaps.

Sites a and b are located in Greenland and Antarctica, respectively, where the polar ice sheets are melting rapidly as the global temperature rises in this century [44]. In Greenland (Figure 4a) and Antarctica (Figure 4b), the EWH series of SSA, IGG, and QF truncated to d/o 12 consistently reflect the long-term losses in the polar ice sheet mass, but IGG shows stronger volatility relative to the other two, especially in Antarctica. The volatility in IGG is more evident when referring to the d/o 60 results. Furthermore, it is shown that all three datasets have the potential to reconstruct the stable seasonal variation in Greenland during the gap months.

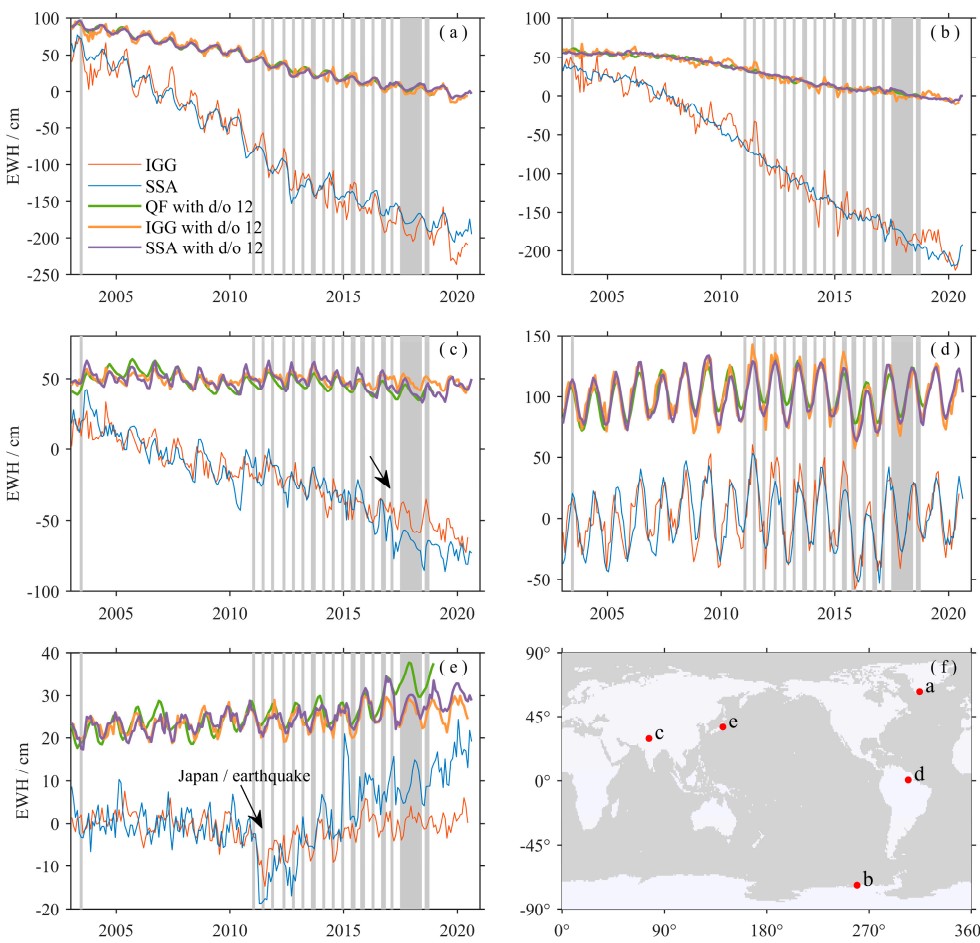

**Figure 4.** EWH time series from five sites in well-studied regions: (**a**) Greenland, (**b**) Antarctica, (**c**) north India, (**d**) Amazon, (**e**) eastern Japan. Subplot (**f**) shows the locations of these five sites. The gray ribbons represent the data gaps.

Sites c and d are located in northern India and the Amazon basin, respectively, two hotspots for research on changes in terrestrial water storage. Figure 4c illustrates the differences in the abilities of these data to monitor the serious groundwater depletion in north India [45]. It is obvious that the EWH results with d/o up to 12 failed to detect this long-term groundwater signal due to the low spatial resolution. The EWH results of IGG and SSA with d/o up to 60 can essentially capture this signal. However, there was a certain difference between SSA and IGG after August 2016, when the declining trend of IGG was significantly smaller than that of SSA. Since IGG did not use GRACE data after August 2016, we speculated that IGG may have underestimated the EWH trend in this region after 2016. The Amazon Basin has the world's greatest variations in water storage due to enormous precipitation, and its interannual variability in water storage is also influenced by El Niño Southern Oscillation (ENSO) events [46]. Figure 4d shows that the EWH series of SSA, IGG, and QF are in good agreement with each other, and all can reflect seasonal and interannual water storage changes in the Amazon Basin.

It has been verified that GRACE is capable of observing the co-seismic and post-seismic signals caused by giant subduction earthquakes [47,48]. Figure 4e illustrates the differences in the abilities of these data to monitor the gravity field signal of the M 9.0 earthquake in eastern Japan on 11 March 2011. It can be found that the EWH results with d/o up to 12 are not capable of monitoring the co-seismic or post-seismic signals. The EWH results of IGG and SSA with d/o up to 60 can essentially reflect the co-seismic signal and the post-seismic signal before August 2016. Furthermore, the SSA data connects the GRACE and GRACE-FO results seamlessly and roughly accurately depicts the gradual and persistent

post-seismic adjustment signal. Unfortunately, the IGG data appears to fail to reflect the persistent post-seismic adjustment signal after August 2016.

Figure 4 shows that the QF, SSA, and IGG data are in high agreement in reflecting the signal of polar ice melting or strong terrestrial water storage variations. The trends of the EWH time series with d/o up to 12 are lower than the results with d/o up to 60, which should be attributable to the fact that only single points were investigated. Limited by spatial resolution, QF is not suitable for monitoring co-seismic or post-seismic signals. Compared to QF and SSA, IGG seems to have a higher signal amplitude. Furthermore, there existed certain differences between SSA and IGG after August 2016 (Figure 4c,e). A possible reason for this discrepancy is that GRACE data after August 2016 was not used when the IGG recovered the time-varying gravity field from the SLR, and the emerging differences likely arise from the divergent spatial resolution between the two kinds of gravity measurements.

### 3.3. Consistency Analysis in the Spatial Domain

Here we scrutinized these three datasets in the spatial domain and evaluate their differences. Similar to Section 3.1, we also compared the three datasets over the three periods: The end of GRACE (January 2016–June 2017), the gap months between the GRACE and GFO missions (July 2017–May 2018), and the beginning of GFO (June 2018–December 2018). Considering that the resolution of QF is approximately 1000–2000 km, we only compared the EWH results for SHCs truncated to degree 12. The GIA effect was also not considered here.

Figure 5 shows the spatial distributions of rRMS among the IGG, SSA, and QF results. Recall that two series are correlated when the rRMS is less than 1.0 and implies a greater consistency. In most regions, the rRMS values of the three datasets are less than 1.0, indicating good consistency. In view of the scarcity of ground-truth data, the measurement-based QF can be treated as an approximately true value during the gap months. According to the statistical analysis in Figure 6, the rRMS values among IGG, SSA, and QF were approximately the same over the three periods, with approximately 60% less than 0.5 and approximately 85% below 1. The consistency of the statistical results further validates the reliability of the three gap-filling datasets. Actually, the rRMS values between SSA and IGG are concentrated in the 0–0.5 range and are slightly higher than those of IQ and QS, indicating a high consistency between SSA and IGG. The main reason may be that both of these two approaches are based on GRACE data.

The SSA, IGG, and QF results are generally consistent but significant differences are found in some areas. Next, we chose six sites with large rRMS values (close to or greater than 2) in Figure 5 and compared their EWH time series. The six sites (a–f) were examined in terms of correlation and RMS of the EWH series in Figure 7. Panels a, b, c, and d of this figure plot the results over the entire period, the end of GRACE, the gaps between GRACE and GFO, and the beginning of GFO, respectively. IGG, SSA, and QF series are well correlated at sites b and d, with all correlation coefficients exceeding 0.76. There is no significant difference between the RMS values of SSA and QF, but the RMS values of IGG are greater than those of SSA and QF, which is particularly evident in the gap months (Figure 7c). That is, the signal recovered from IGG might be higher than from SSA and QF, but there is a good agreement in the seasonal variation of QF, SSA, and IGG. It can also be concluded that the correlation between QF and SSA is greater than the correlation between QF and IGG, except for sites e and f. Considering that QF is purely derived from satellite observations, we find that SSA shows better agreement with the observation-based results than IGG. In contrast, at sites with small RMS values, which means weak hydrological signals, such as those less than 5 cm, there was a large difference between the three datasets. Both the RMS and correlation coefficients are divergent, especially in sites a and e. Thus, we suggest that these three gap-filling data should be cautiously used in regions with weak hydrological signals.

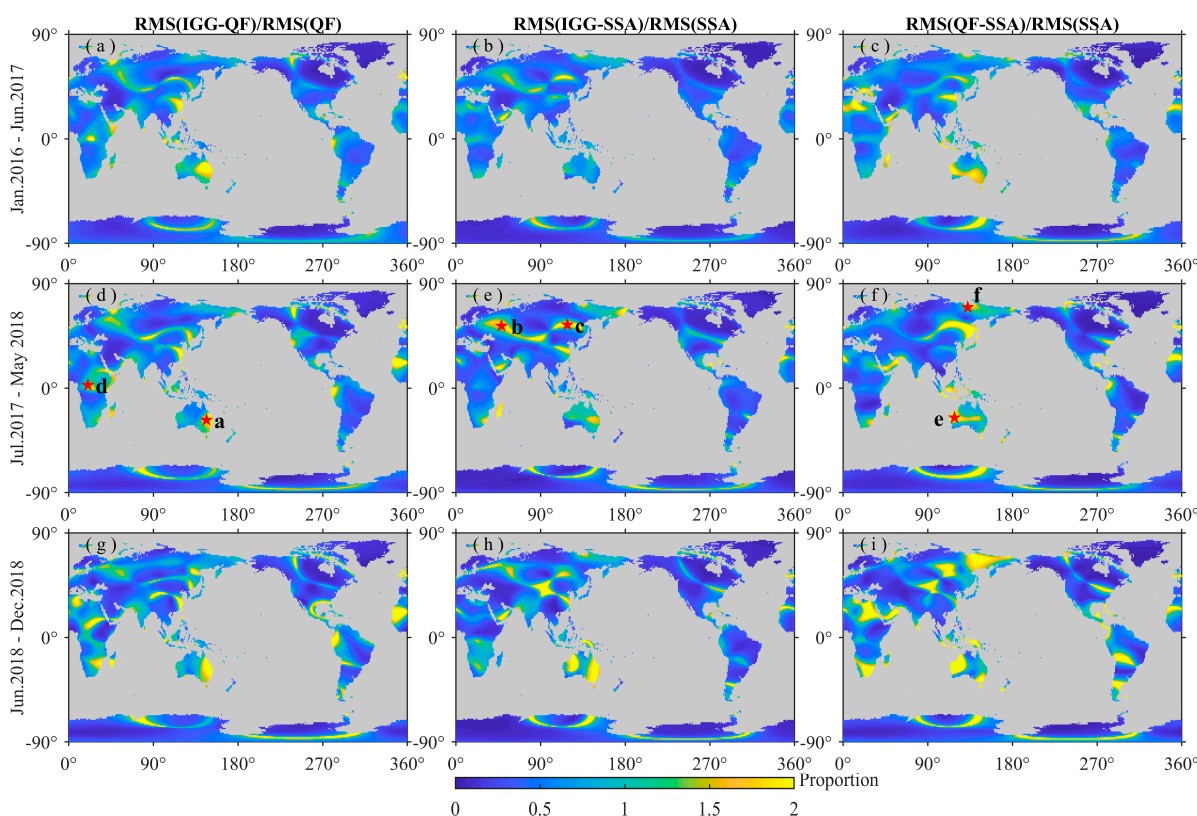

**Figure 5.** Worldwide comparison of the SSA, IGG, and QF results during (**a–c**) January 2016 to June 2017, (**d–f**) July 2017 to May 2018, and (**g–i**) June 2018 to December 2018. Plotted are the rRMS values of IGG relative to QF (subplots (**a,d,g**)), IGG relative to SSA (subplots (**b,e,h**)) and QF relative to SSA (subplots (**c,f,i**)).

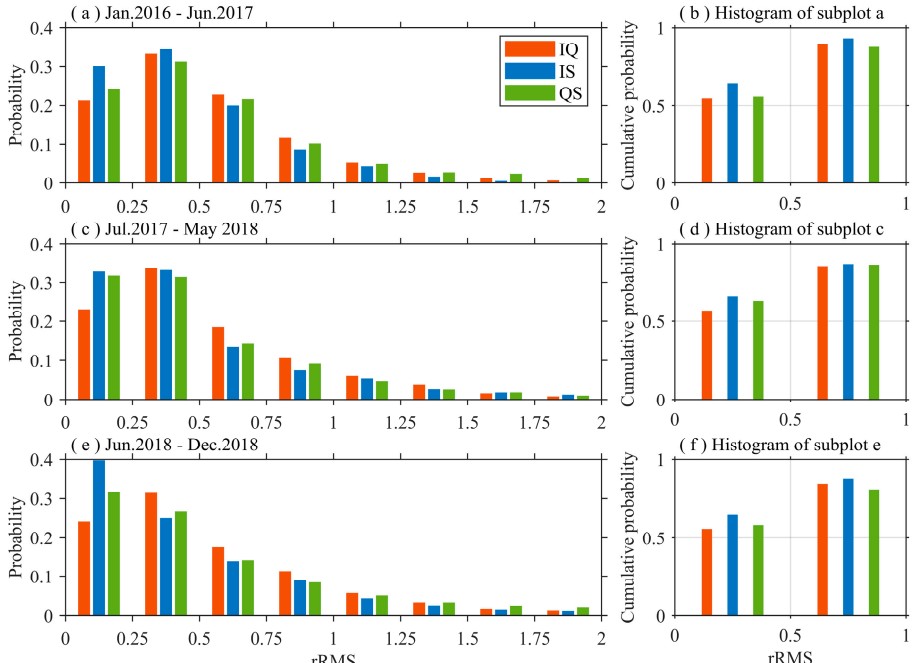

**Figure 6.** Probability distributions of Figure 4. Subplots (**a,c,e**) display the probabilities of rRMS values of IGG relative to QF (IQ), IGG relative to SSA (IS), and QF relative to SSA (QS), respectively, during three periods: January 2016 to June 2017, July 2017 to May 2018, June 2018 to December 2018. Subplots (**b,d,f**) display the cumulative probabilities of (**a,c,e**), respectively.

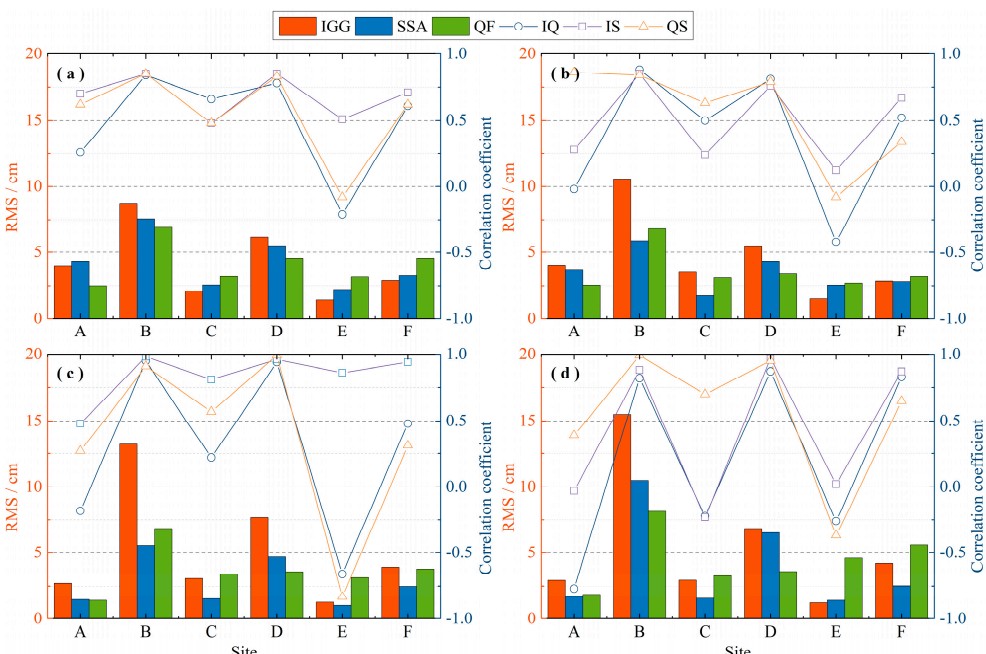

**Figure 7.** Comparison of correlations and RMS values at six sites with large differences in their rRMS values (see Figure 6 for locations). The analysis was performed over four time periods: (**a**) January 2003 to December 2018; (**b**) January 2016 to June 2017; (**c**) July 2017 to May 2018; (**d**) June 2018 to December 2018. IQ, IS, and QS represent the correlations between IGG and QF, IGG and SSA, and QF and SSA time series, respectively. The bars and series represent the RMS values and correlation coefficients, respectively.

In summary, in most regions, the global terrestrial EWH results based on the degree-12 SHCs are highly consistent. More than 81% of the rRMS values estimated between any two of the three datasets are less than 1.0, with the ratio reaching 94% in some cases. The datasets are strongly correlated when larger RMS values (e.g., >5 cm) exist in the EWH series, suggesting good agreement on the seasonal patterns of variation. However, the gap-filling signals recovered from IGG might be higher than those recovered from SSA and QF, which was inferred similarly in Section 3.2. In contrast, the RMS and correlation coefficients are heterogeneous at sites with weak hydrological signals, showing poor consistency. As a result, we recommend that these three gap-filling datasets be utilized with caution in regions with weak hydrological signals.

### 3.4. Global Mean Sea Level Change

The global sea level change (measured by satellite altimetry) consists of the changes in sea mass (measured by GRACE) and density (i.e., steric change, measured by Argo floats). The consistencies among these three observations provide a possible way to evaluate the data quality of the GRACE gap filling products, since the other two datasets do not suffer from discontinuities. Figure 8 compares the time series of the global mean sea level anomalies calculated from the IGG and SSA data. The altimetry data, provided by the Commonwealth Scientific and Industrial Research Organization (CSIRO, http://www.cmar.csiro.au/sealevel/sl_data_cmar.html, Canberra, Australia, April 2022), combine the data of TOPEX/Poseidon, Jason-1, Jason-2, and Jason-3 and are corrected for the inverse barometer and GIA effects. The monthly gridded 1° × 1° Argo data were provided by the Scripps Institution of Oceanography [49]. The GIA effect in the QF, IGG, and SSA observations was also corrected using a three-dimensional GIA model [50]. The ocean-atmosphere model (GAD) was reintroduced into the GRACE SHCs, and the time-varying component of the monthly average atmospheric mass over the ocean was eliminated to account for the inverse barometer effect. Furthermore, all the time series were constructed using an ocean mask, which excludes coastal areas within 300 km. To calculate the ocean

water mass, QF was truncated to degree 12 (Section 3.1), but the full spectrum (degree 60) of IGG and SSA was used.

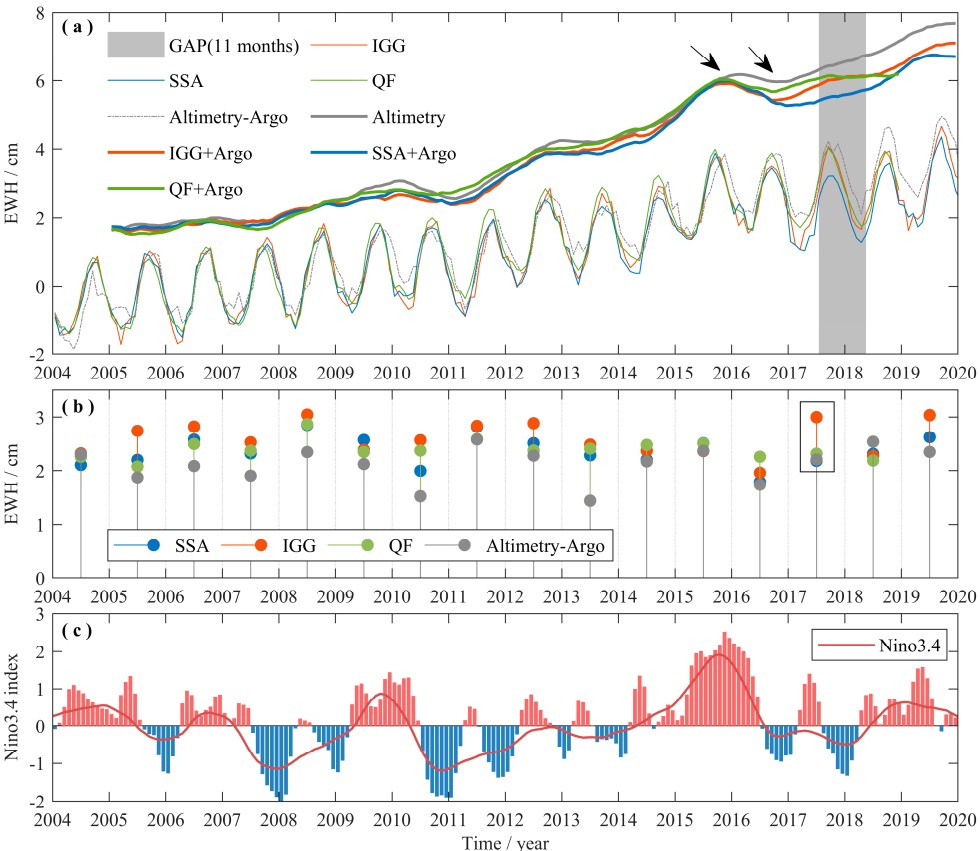

**Figure 8.** Comparisons of time series of global sea level anomalies. (**a**) Global mean sea level anomalies. QF, IGG, and SSA represent the seawater mass changes and Altimetry-Argo is the altimetry result minus the steric sea level change (note that seasonal signals remain in these data). Altimetry, SSA + Argo, IGG + Argo, and QF + Argo denote the corresponding total sea level changes after 12-month sliding average. (**b**) Annual variations in mean seawater mass. (**c**) Time series of Nino3.4 indicators. The bars and curves represent the monthly values and the series after 12-month sliding average, respectively.

Figure 8a shows the seawater mass changes estimated from QF, IGG, and SSA, and the mass changes derived from the difference between Altimetry and Argo. The series is highly consistent over the entire period (January 2004 to December 2019). The correlations among QF, IGG, and SSA are all above 0.95, and the rRMS values between the three are all below 0.5. For instance, the rRMS value of IGG relative to SSA is just 0.15. After 2016, the seasonal variations in the three series remained consistent but they gradually departed from the Altimeter-Argo series. Notably, the SSA and IGG estimates of seawater quality were significantly lower than the Altimetry-Argo estimates at the beginning of 2017, but then there was an anomalous increase in IGG that mostly compensated for this difference, resulting in a great seasonal variation in 2017. In terms of annual variation, IGG results tend to be higher than SSA, QF, and Altimeter-Argo (Figure 8b). This difference is particularly significant in the 11-month gap, where the difference between IGG and SSA reached 7.8 mm. In contrast, the SSA results are generally consistent with the annual variation of Altimetry-Argo.

Next, we scrutinized the discrepancies after 2016. The interannual variations were obtained by averaging the series over a sliding 12-month window (Figure 8a) and they were shifted upwards by 2 cm for better visualization. Figure 8a shows that all the IGG + Argo, SSA + Argo, and IGG + Argo series fit well with the altimetry series before August 2016.

However, the four series exhibited substantial discrepancies after 2016, with lower values for SSA + Argo, IGG + Argo, and QF + Argo than the altimetric results. There are several possible causes: (a) Deteriorated data quality during the late stages of GRACE satellites operations and the reduced number of accelerometers after 2015 [51,52]; (b) inaccuracies of Argo data [53]; and (c) uncertainties in the estimated degree-1 series [51]. Although these large discrepancies prevent a quality evaluation based on the sea level budget approach, the IGG + Argo and QF + Argo series consistently show closer agreement with Altimetry than SSA + Argo during the gap period, indicating that SSA may underestimate the mean seawater mass variation in the gap months. However, it is important to note that the IGG series suspiciously shows a strong seasonal increase in 2017 (marked by the box in Figure 8b) when the ENSO index is weak (Figure 8c). According to previous records, strong seasonal changes always occur when the ENSO index is extremely negative (e.g., in 2008, 2011, and 2012).

In conclusion, Figure 8 shows that the IGG, SSA, and QF estimates of global ocean water mass began to differ from the Altimetry-Argo result toward the end of the GRACE mission. IGG appears to overestimate the annual amplitude of seawater mass, but the inter-annual mean sea level change derived by it is largely consistent with QF. SSA seems to underestimate the trend during the gap period. The cause of the large bias in the sea level budget after 2016 is unclear, and these data require further examination when used to analyze the global mean sea level changes in recent years.

### 3.5. Comparison with Water Storage Model

Here, we adopted a climate-driven water storage change model (GRACE-REC) provided by Humphrey and Gudmundsson (2019) [54] to test the ability of the gap-filling products to recover hydrological signals. Due to the low resolution of the QF dataset, we ignored it in our analysis here.

Figure 9 compares the RMS values of residuals of IGG and SSA by subtracting the GRACE-REC model for results between July 2017 and May 2018. The difference between the gap-filling data and the model values may include unmodeled signals and the model deficiency (which is assumed to be small). Figure 9a,b shows the spatial distribution of the RMS values, where values less than 1 indicate two datasets are in some way consistent. As shown in Figure 10d, SSA has a higher proportion of relative RMS values smaller than 1 than IGG (65% versus 59%). Moreover, the spatial distribution results in Figure 9a,b show that the IGG was significantly wider than the SSA for regions with relative RMS values greater than 1. Therefore, we conclude that the SSA gap-filling series is more consistent with GRACE-REC than with IGG and shows better performance in restoring transient hydrological events.

Figure 10 shows the time series of EWH and precipitation at points A, B, C, and D (marked in Figure 9), where the relative RMS values are large, to illustrate the potential bias in IGG and SSA. The precipitation data are the monthly average reanalysis data, expressed as EWH, obtained from the Climate Data Store (https://cds.climate.copernicus.eu/#!/home, Brussels, Belgium, April 2022). The water storage in point A shows a 2010 drought and a post-2016 increase, which are consistent with the progression of dry and wet periods. The alternation of positive and negative water storage anomalies is well-reflected by the SSA and GRACE-REC results, but not in the IGG results. This case demonstrates that using the first six modes in IGG may lose some interannual variation in regional signals. This problem can also be found at point C, where the IGG result fails to capture the water storage after 2016 due to the contemporaneous increase in precipitation. The time series of IGG and SSA at points B, C, and D are fairly consistent until 2017, after which their differences start to increase. The main reason for this discrepancy may be that the IGG only used GRACE data from February 2002 to August 2016, but the extrapolated gravity fields from the SLR could not maintain the previous variation pattern and may not accurately reflect some sudden mass change signals as GRACE does. The series of SSA and GRACE-REC were mostly consistent, especially before and after the gap period. However,

the SSA data have one big shortcoming. As it is totally data-driven, it cannot capture a suddenly emergent climatic event during the data gap. For instance, point D located in North America was influenced by a sudden precipitation deficit in 2016, resulting in GRACE-REC results showing a sharp decrease in water storage. However, SSA reflects only a moderate decrease in water storage in 2016. Nevertheless, such interannual variability is absent in the IGG result as well.

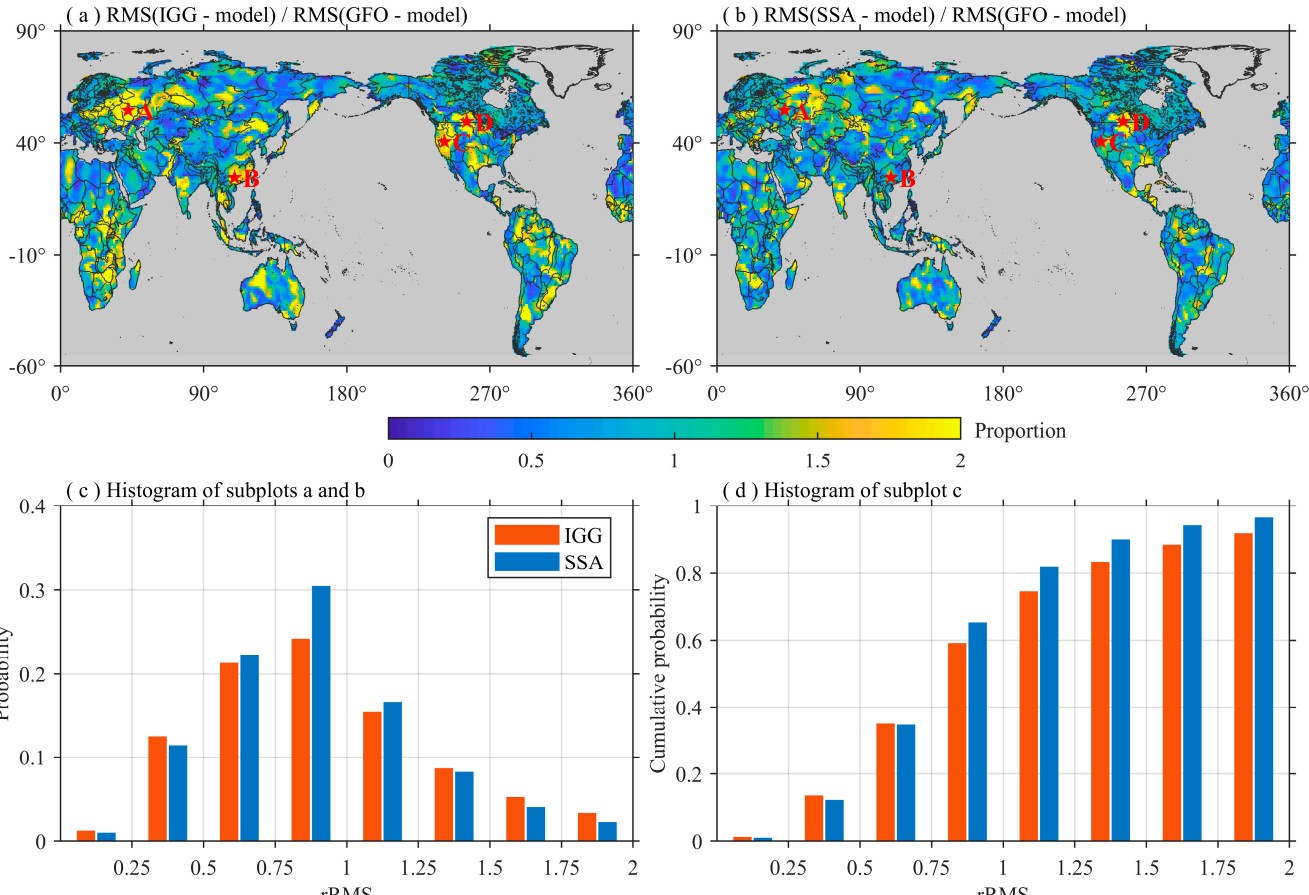

**Figure 9.** Comparison of the RMS values of residuals of (**a**) IGG and (**b**) SSA by subtracting GRACE-REC. (**c**) Probability and (**d**) cumulative distribution of grid values in (**a**,**b**).

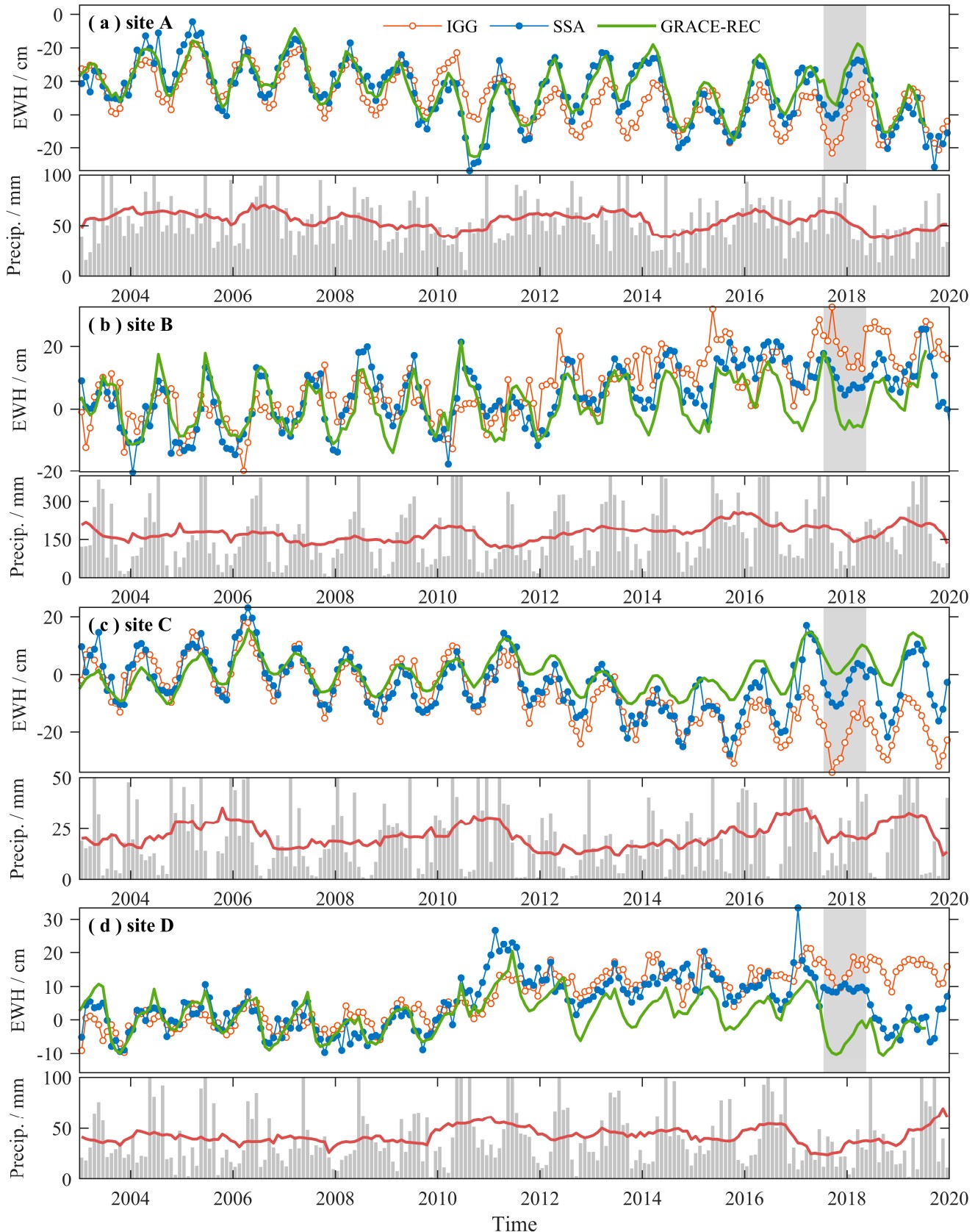

**Figure 10.** Time series of EWH and precipitation at points A, B, C, and D (marked in Figure 9): (**a**) point A, (**b**) point B, (**c**) point C, (**d**) point D. The red curve in the precipitation plots is the smoothed result of a 12-month sliding window applied to the monthly precipitation (gray bars).

## 4. Triple-Collocation Method-Based Fusion on the Gap-Filling Data

Here, we put forward a new strategy based on the triple-collocation method to generate a fused SHC gap filling product, which theoretically avoids random error better than any of the individual datasets. We first estimated the error variance of each degree and order based on the 10-year SHCs data from 2009 to 2018 and counted the median variance of each order. Note that the calculated error variance may occasionally be negative (less than 25%) due to increased correlation among the series, which will affect the calculation of weights. Therefore, to get rid of these negative values, we use the median error variance of each degree (l) to calculate the weight, which is used for all coefficients of the same degree (i.e., we assume the data quality is a function of only degree). Regarding the finite spatial resolution of the QF, only the low-degree terms (degree 2–15) were fused. Figure 11 shows the distribution of weight values for each degree of SHCs. Compared to IGG and SSA, the weights of QF are lower and gradually decline with an increasing degree. This result is consistent with our previous finding that the higher degree terms of the SHCs of QF may contain higher errors and should therefore be given smaller weights when performing the data fusion.

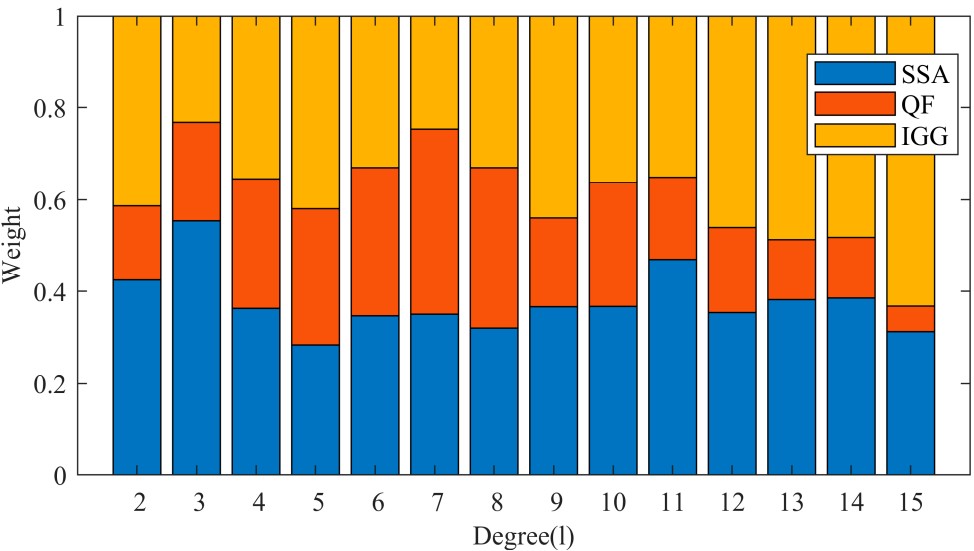

**Figure 11.** Distribution of weight values for fusing the three SHC datasets used in this study.

Next, the weights calculated in Figure 11 were applied to fuse the SHCs of SSA, QF, and IGG. We only show the results of coefficients of order = 1 in Figure 12. The weighted average results retain the interannual signal to a large extent and provide better agreement with the IGG and SSA series and effectively avoid the detrimental effects of systematic bias in QF, such as $C_{2,1}$, $C_{5,1}$, $C_{13,1}$, and $C_{15,1}$. The fused series shows a large difference when the individual series are subject to high-frequency vibrations, such as $C_{5,1}$, $C_{7,1}$, $C_{13,1}$, and $C_{15,1}$, and the weighted average is evidently smoother, indicating the noise is suppressed to a certain extent. In some cases, the fused series can better maintain the seasonal and interannual variation patterns in the gap months. For example, the agreement for the results of $C_{7,1}$ is poor: IGG and QF appear to over- or underestimate the value of the gap period, and SSA, while retaining the long-term trend, loses much of the seasonality (Figure 12f). Nevertheless, the fused series retains both the long-term trend and part of the seasonal signal.

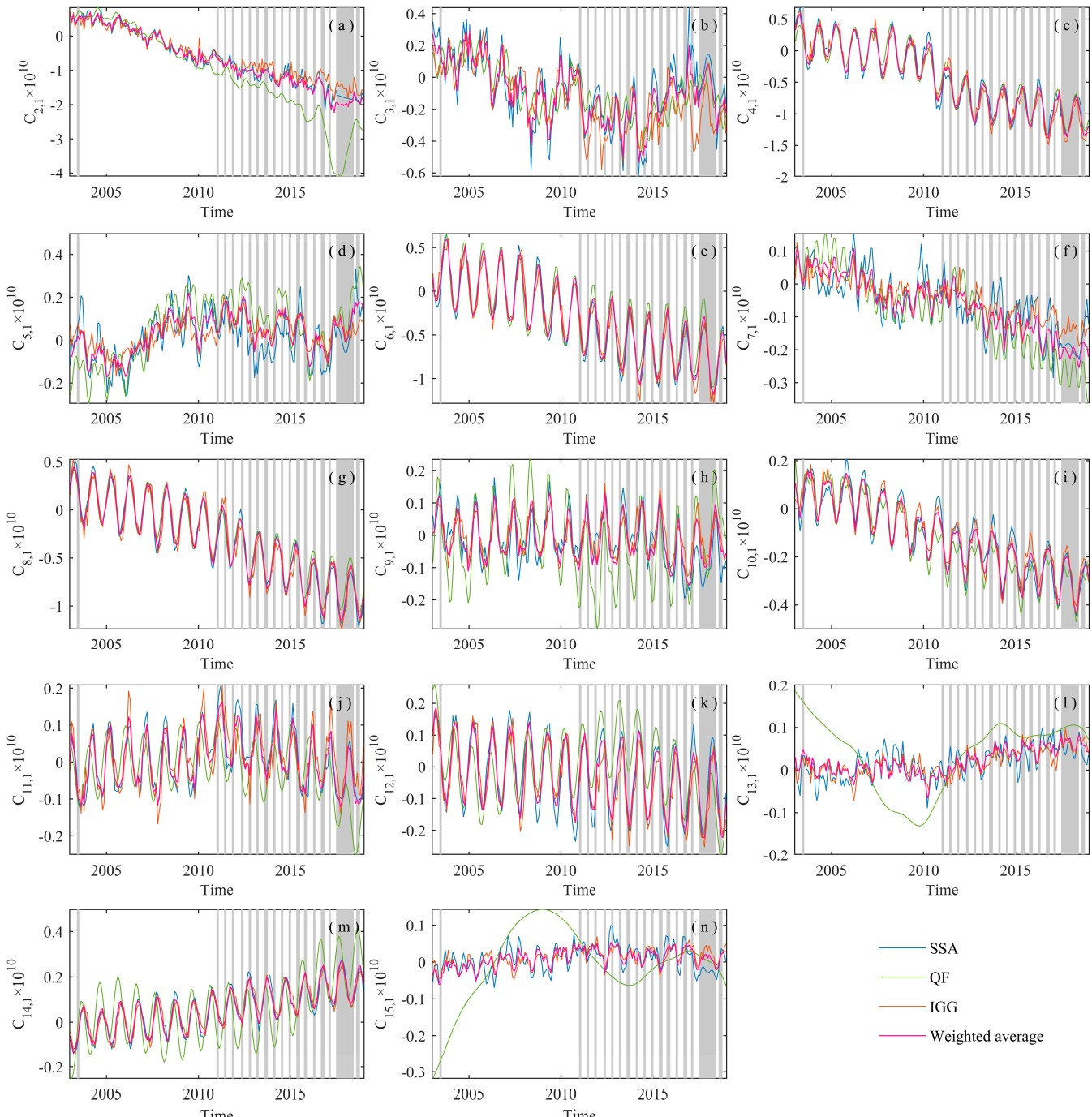

**Figure 12.** Time series of selected coefficients of the SSA, QF, IGG, and the fused results. The gray bars indicate the location of the missing data of GRACE. (**a–n**) denote the $C_{2,1}$ to $C_{15,1}$ time series, respectively.

## 5. Discussion on Dataset Selection

There were three SHC gap-filling products chosen here for systematic analysis and comparison. QF is derived from HLSST tracking and SLR data, which can be considered real observations. When there are no other data available, it can be a good candidate to fill the gaps between the GRACE and GFO periods. However, due to the limitation of the observation technology, the effective resolution of QF is currently limited to a maximum of ~1000 km, which corresponds to the SHC degree 20 [31,55]. IGG is derived from SLR data using the GRACE empirical orthogonal function decomposition model, which can estimate the same spatial resolution gravity fields as GRACE [16]. Considering only GRACE data

prior to August 2016 are used in IGG and only six EOFs for GRACE are used at most, it may result in partial loss of actual signals and increasing uncertainties after August 2016. SSA is a purely data-driven approach fully inherited from GRACE/GFO observations, so it is easy to implement and can maintain the spatial resolution of GRACE.

Since these three types of data are produced by different approaches, they may be applicable to different application fields. Obviously, we cannot simply determine which data are better or worse but need to analyze the data in a comprehensive manner. As mentioned in Section 3, we focused on several topical issues: Changes in terrestrial water storage, polar ice melting, global mean sea level change, the co-seismic and post-seismic signals, and sudden hydrological events. First, all these data have the ability to monitor changes in strong terrestrial water storage (Figure 4d) and polar ice melting (Figure 4a,b) but should be utilized with caution in regions with weak hydrological signals (Figure 7). Second, QF is suitable only for large-scale mass migration studies due to significant noise in high-order terms (Figure 3), but it can be a good candidate to fill the gaps between GRACE and GFO if no other data are available. Third, since only GRACE data prior to August 2016 were used in IGG [16], IGG may not accurately capture some sudden mass-changing signals after August 2016. Therefore, SSA is more apt at producing GRACE-like results when monitoring these signals after 2017 (Figure 4c,e). Fourth, as the IGG data use only six EOFs for GRACE at most, they may occasionally miss the inter-annual variation in local areas. So SSA may be more reliable to monitor hydrological signals (Figure 10). However, it should be noted that SSA is entirely data-driven rather than actually observed [21], and it occasionally fails to capture sudden climatic events during data gaps (Figure 10d), so we need to use these data with caution as well.

To reduce the frequency of random errors and outliers in the gap-filling product, we proposed a new scheme to fuse these three datasets based on the TC method. The fusion results suppress the signal noise while maintaining the seasonal and interannual variation patterns in the gap months. The fusion results may not always be the best, but they ensure consistently acceptable performance. Therefore, when it is difficult to determine which data are more appropriate, we recommend the easy-to-implement fusion results.

## 6. Conclusions

The 11-month gap between the two missions inevitably limits our ability to systematically analyze and fully utilize the satellite observations of GRACE and GFO. In this paper, we scrutinized three gap-filling datasets. To verify the differences and applicability of the three types of gap-filling methods, we analyzed and compared the QF, IGG, and SSA data in the spectral and space domains. The main conclusions of the study are summarized below.

(1) The SHCs of the QF, IGG, and SSA data are consistent up to degree 12. On the one hand, the SHCs of these three datasets show good consistency, with more than 80% of rRMS values <1 relative to Swarm. On the other hand, the global terrestrial EWH results based on the 12-order SHCs are still consistent in most parts of the world, especially in reflecting the signals of polar ice melting and strong terrestrial water storage variations. It is noticeable that there is better consistency between IGG and SSA due to the fact that both SSA and IGG data are highly inherited from GRACE/GFO.

(2) The IGG and SSA data are basically consistent over the 11 gap months, with the signal intensity in IGG slightly higher than in SSA, but the IGG shows a faster increase in the mean ocean water mass and the SSA appears to better capture the interannual variation in the terrestrial water storage.

(3) The major shortcomings of IGG data are mainly reflected in two aspects. First, as the IGG data use only six EOFs for GRACE at most, it may occasionally miss the inter-annual variation in local areas or fail to detect post-seismic adjustments of giant earthquakes. Second, the IGG recovers the time-varying gravity fields from the SLR using only GRACE data prior to August 2016, which causes the IGG to maintain the previous variation pattern during the extrapolation time period, so it may not accurately capture some sudden mass-changing signals after August 2016.

(4) The lower-degree terms of the QF coefficients ($l \leq 16$) are essentially consistent with those of SSA and IGG, but the noise increases significantly for the high-order terms of the QF data ($l > 16$), so this solution is suitable only for large-scale mass migration studies.

(5) Based on the triple collocation method, we propose a new scheme to derive a weight matrix that can fuse these three datasets into a more robust solution. Limited by the resolution of QF data, here we only fused the SHCs data up to degree 15, but they can be extended to higher degrees should higher-resolution data be released in the future.

There are other uncertainties in the latest gap-filling data that should be noted. First, the gap-filling data should be carefully interpreted during transient climatic events. Either IGG or SSA may fail to capture changes in water storage caused by such events. Second, these three gap-filling datasets should be utilized with caution in regions with weak hydrological signals due to poor consistency in RMS and correlation coefficients. Third, due to the unexplained large discrepancies among the altimeter, Argo, and GRACE data, the filling data require careful interpretation when used to monitor the global mean sea level changes after 2016.

**Author Contributions:** Conceptualization, A.Q. and S.Y.; methodology, A.Q. and S.Y.; formal analysis, A.Q., F.L., B.S., and G.S.; investigation, A.Q. and S.Y.; writing—original draft preparation, A.Q., F.L., B.S., G.S., and X.L.; writing—review and editing, S.Y.; visualization, A.Q.; supervision, S.Y.; project administration, S.Y.; funding acquisition, A.Q., S.Y., F.L., G.S., and B.S. All authors have read and agreed to the published version of the manuscript.

**Funding:** This research was funded by the Hebei Key Laboratory of Earthquake Dynamics Open Fund (grant no. FZ202214), the University of Chinese Academy of Sciences Research Start-up Grant (grant no. 110400M003), the Self-Funded Project of Scientific Research and Development Plan of Langfang Science and Technology Bureau (grant nos. 2020013045 and 2021011030), the Central University Basic Research Fund of China (grant nos. 2020013045 and ZY20215159), the Key Project of Science and Technology Research for Universities of Hebei Province (grant no. ZD2020407), the Hebei Key Laboratory of Earthquake Disaster Prevention and Risk Assessment (grant no. FZ213109), and the Beijing Key Laboratory of Urban Spatial Information Engineering (grant no. 20220102).

**Data Availability Statement:** The SSA data are available at https://doi.org/10.18419/darus-807. The IGG data are available at https://doi.org/10.22000/357. The QF and Swarm data are available at the International Center for Global Earth Models (http://icgem.gfz-potsdam.de). The altimetry data are available at CSIRO (http://www.cmar.csiro.au/sealevel/sl_data_cmar.html). The Argo data are available at https://argo.ucsd.edu/data/argo-data-products/. The precipitation data are obtained from Climate Data Store (https://cds.climate.copernicus.eu/#!/home). The climate-driven water storage changes model (GRACE-REC) is available for download at the website https://figshare.com/articles/dataset/GRACE-REC_A_reconstruction_of_climate-driven_water_storage_changes_over_the_last_century/7670849.

**Acknowledgments:** We are grateful to Löcher and Kusche for providing IGG data. We thank the International Centre for Global Earth Models for providing QF data. Thanks to CSIRO for providing sea level height data for "Combined TOPEX/Poseidon, Jason-1, Jason-2/OSTM and now Jason-3 sea level fields – several versions". Special thanks to the Commonwealth Scientific and Industrial Research Organization for providing altimetry data. Furthermore, thanks to the Scripps Institution of Oceanography for providing the monthly gridded Argo data.

**Conflicts of Interest:** The authors declare no conflict of interest.

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
