# Peer review of "Evaluation of the Consistency of Three GRACE Gap-Filling Data"

_remotesensing, doi:10.3390/rs14163916_

Round 1

Reviewer 1 Report

Review of “Evaluation of the Consistency of Three GRACE Gap-Filling Data” by Qian et al. (remotesensing-1845663)

This paper focuses on three typical products (QF, SSA, IGG) for filling missing data between GRACE and its Follow-on. The authors have done a lot of work to show the consistency of these three products and their respective applicability, and based on the three data fusion to obtain a more stable dataset. These works are of great help for subsequent researchers to select suitable datasets. On the whole this paper should be some minor revision before final approval. My other comments are below.

(1) The author has emphasized several times that the spatial resolution of the QF dataset is much lower than that of GRACE, which makes it difficult for this dataset to have good consistency with the other two GRACE-based datasets (IGG, SSA). For example: Page6-Line228-229, Page7-Line261-263, Page11-Line 345-346 and other places describe the comparison results. I think authors should choose data with more consistent spatial and temporal resolutions as much as possible when comparing multiple datasets, otherwise the credibility of the comparison results will be reduced.

(2) When comparing the EWH results in the spatial domain, the IGG dataset is in good agreement with the SSA. However, the authors have repeatedly emphasized that since the results of the IGG data recovery after August 2016 did not use the GRACE data, the results after this time were not consistent with the SSA. Here, the authors need to clarify why there is such a difference in the results recovered without GRACE data, and which dataset is more trustworthy compared to SSA (Page10-Line307-311, Page10-Line333-335).

(3) In this paper, the authors focus more on the performance of the three datasets in various comparisons, but less on the specific reasons for these performances. The reader or user may be more concerned about why these performances make the dataset more credible, in other words, the main benchmarks considered when choosing the data, etc.

Reviewer 2 Report

THis is a very thorough comparison of three approaches to fill the gap between GRACE and GRACE FO data. THe manuscript is clear, thorough, well written, the figures are clear and necessary, and I could not detect any errors.

A couple minor edits:

L 111: comparation -> comparison

L 425: Figure 9a -> Figure 8a

L517: triple collation -> triple collocation

L 584: should higher resolution data is released -> if higher resolution data is released (or should higher resolution data be released)
